# Development and validation of the MY-VEG-FFQ: A modular web-based food-frequency questionnaire for vegetarians and vegans

**Kerem Avital**[1], **Sigal Tepper**[2], **Sivan Ben-Avraham**[1], **Danit Rivka Shahar**[1] *

**1** The International Center for Health Innovation & Nutrition, School of Public Health, Faculty of Health Sciences, Ben-Gurion University of the Negev, Beer-Sheva, Israel, **2** Department of Nutritional Sciences, Tel-Hai College, Tel Hai, Israel

* dshahar@bgu.ac.il

**Data Availability Statement:** All relevant data are within the manuscript and its Supporting Information files

## Abstract

### Background and objective

The adoption of plant-based diets in recent years has increased the need for accurate assessments of dietary intake among vegans, vegetarians, semi-vegetarians, and omnivores. This study aimed at developing and validating a modular web-based food-frequency questionnaire (FFQ), the MY-VEG-FFQ. This FFQ was based on the original FFQ (O-FFQ) designed for the Israeli population and incorporates a skip algorithm tailored for different dietary patterns.

### Methods

A convenience sample of 101 participants, recruited via social media, completed the MY-VEG FFQ, as well as a three-day food records, which served as the gold standard for this research. Relative validity of the new FFQ was evaluated by comparing nutrients with those in the three-day food records, using Pearson correlation coefficients, Bland-Altman plots, and cross-classification. The results were compared with 90 O-FFQs that previously had been completed by vegans.

### Results

The validation analysis showed that nutrient-intake estimates were generally higher for the MY-VEG-FFQ than those of the three-day food records. Pearson correlation coefficients ranged between 0.25–0.63, indicating an acceptable agreement between the two tools. The proportion of participants with exact or adjacent quartile agreement was between 73%–82%. The Bland-Altman analysis revealed overestimation of nutrient intake via the MY-VEG-FFQ. Compared to the O-FFQ, vegans who completed the MY-VEG-FFQ reported consumption of more food items. Additionally, the MY-VEG-FFQ showed a significantly higher intake of most macro- and micronutrients.

**Funding:** The authors received no specific funding for this work.

**Competing interests:** The authors have declared that no competing interests exist.

## Conclusions

The My-VEG-FFQ demonstrated reasonable validity in assessing dietary intake among people who followed a plant-based diet. However, it tended to overestimate nutrient intake compared to the three-day food records. The development of a modular web-based FFQ with a skip algorithm tailored for specific dietary patterns, fills a crucial gap in accurately assessing the dietary intake of these populations. The MY-VEG-FFQ offers a practical and cost-effective tool for evaluating long-term dietary consumption among people who follow different dietary patterns.

## Introduction

In recent years, primarily due to health, ethical, and sustainability concerns, an increasing proportion of the global population has begun adhering to a plant-based diet, either partially or in full [1]. Such diets are perceived as beneficial to human health [2] and to the environment [3]. Based on surveys and online polls, it seems that veganism has become much more globally prevalent, with estimations indicating that a vegan diet is followed by 4% of the population in Europe, 6% in the USA, and 13% in Asia [4].

Yet as this global prevalence of vegetarians and vegans increases, so do the challenges entailed in accurately assessing their dietary intake. First, both vegetarians and vegans avoid a significant number of foods, consuming instead a range of unique products and adapted dishes that exclude some or all types of animal products [5]. Moreover, data with an emphasis on changes that have taken place over the past decade are greatly lacking regarding the current dietary patterns of vegetarians and vegans. For example, the introduction of plant-based ultra-processed foods, has increasingly replaced homemade food items [6, 7]. A recent analysis by Gallagher et al. highlights the need to investigate different vegan dietary patterns, especially newly emerging ones that differ greatly from traditional vegan diets [8].

One practical tool for assessing long-term dietary intake is the food-frequency questionnaire (FFQ), which ranks food and nutrient intake into high and low frequency of consumption. This tool has been shown to have good validity for ranking nutrient intake [9] and for calculating relative risks of negative health outcomes (such as cardiovascular disease and diabetes) [10]. The FFQ is considered both simple and cost effective [11], especially in its more recent web-based format, which can be completed using most mobile devices. Completing digital questionnaires eliminates issues, such as participants skipping questions or choosing multiple answers for a given question [12].

Web-based questionnaires also enable skip algorithms to present a modularly adjusted list of food items in line with preliminary screening questions regarding dietary habits [13, 14]. For example, if a person defines his/her dietary pattern as vegetarian, then certain items are automatically removed from the questionnaire, while others items are expanded to include dishes and food items that tend to be exclusively consumed by vegetarians and vegans. Omnivorous populations can also use these FFQ tools, thereby enabling important comparisons between populations with different dietary patterns.

The aim of this research was to develop and validate a modular web-based FFQ for all dietary populations (vegetarians, vegans, semi-vegetarians, and omnivores) based on a unique group-tailored skip algorithm.

## Methods

### Study design and participants

First, we developed the new MY-VEG-FFQ (detailed below) based on the previously validated FFQ (O-FFQ). The aim was to offer a modular dietary assessment tool for the entire Israeli population, regardless of their dietary lifestyles. Next, for the validation stage, we recruited 242 individuals, aged 18–68 years, via social media from October 1, 2020 to August 31, 2021. The inclusion criteria required participants to be age 18 or older, self-report as adhering to a vegetarian or vegan diet, and proficient in Hebrew. The participants provided some sociodemographic and lifestyle information, such as age, education level, stature, body mass (for calculating body mass index [BMI]), and smoking habits. We then asked the participants to complete the newly developed MY-VEG-FFQ and submit a three-day food record) 3-DFR (to be used as the benchmark for this study.

The questionnaire offered a modular dietary-assessment method that could be used for the entire population, regardless of their eating habits. To assess the performance of the novel MY-VEG-FFQ, we compared the current findings with the input provided by 90 vegan participants in a previous study. This entailed a cross-sectional population survey (the Tel Hai Cohort) led by our group from February 1, 2021 to March 31, 2022, using the original FFQ (O-FFQ), which had been developed and validated for the general Israeli population [15]. Of the 1,447 participants in this cohort, the 90 vegans became the reference group for the current study. All contributors in both studies provided a written web-based informed consent form prior to completing the questionnaire and submitting additional data via a web-based survey. The Ethics Committees of the Ben Gurion University of the Negev approved this study (#24–2020) as did Tel Hai College (#12/2020-5). All questionnaires were in Hebrew.

### Developing the MY-VEG-FFQ

We developed the MY-VEG FFQ based on the O-FFQ, which was developed and validated by the Nutritional Center at the Ben-Gurion University of the Negev specifically for the Israeli population (the detailed process is described elsewhere [15]). To this core, we added a new range of vegan and vegetarian dishes and foods. Specifically, we built the MY-VEG FFQ as detailed below:

We first created a list of vegan food items, based on 200 anonymous dietary reports submitted to the researchers by four dietitians who primarily worked with a vegan population. Conceptually similar food items, such as lentil soup and bean soup, were aggregated, resulting in a total of 161 items. Next, each item was classified into one of the following three categories: (1) Food items that were identical to those found in the O-FFQ (e.g., rice and tomatoes); (2) Food items included in the O-FFQ that required minor adjustments (e.g., baked goods that traditionally contained dairy or meat); and (3) Completely new food items (e.g., tofu-based dishes). Based on this classification, we created a comprehensive list of foods consumed by vegetarians and vegans. The list was then modified based on the comments of the four dieticians. The questionnaire was adapted to a web-based application, using the LimeSurvey™ open-source software (Hamburg, Germany) that was presented at the authors' affiliated academic institution. When combined, the original and the novel questionnaires revealed a total of 179 food items. To make the questionnaire less burdensome, we used a skip algorithm to create a semi-personalized FFQ, using branching questions about habitual consumption of the following food groups: meat, fish, eggs, dairy products, meat substitutes, and dairy substitutes. Consequently, each participant received a different list of items based on the branch questions (S1 Fig).

Similar to the O-FFQ, each food item was presented with a portion size (using household measurements or weight). Participants were asked to rate the frequency in which they consumed that specific item, as per the following scale: (a) Never or less than once per month (b) 1–3 times per month; (c) 1–2 times per week; (d) 3–4 times per week; (e) 5–6 times per week; (f) once per day; 2–3 times per day; (g) 3–5 times per day; or (h) 6 times per day or more. Respondents could only mark one answer for each item. Explanations regarding the serving size were given for each question. An English translation of the MY-VEG-FFQ is provided in the S1 File. In the final stage of developing the questionnaire, a validation study was performed among vegetarians, vegans, and semi-vegetarians (the latter reporting that they eat meat and fish less than once a week).

At this stage, the participants completed the new MY-VEG-FFQ and submitted a hard copy or web-based 3-DFR (covering two weekdays and one weekend day). We then analyzed the reported food nutrients using *Tzameret* software, developed by the Israel Ministry of Health for food analysis [16]. Next, we documented and analyzed these submitted records. Food items that were missing from the *Tzameret* software were added manually, using data from the United States Department of Agriculture) USDA (database [17], the manufacturer, or a combination of the two. We also entered new recipes for dishes consumed by vegan participants.

The average food consumption per day of each participant, as seen in the 3-DFR, was calculated as follows:

$$\frac{WE*2 + WD_{day1}*\frac{5}{2} + WD_{day2}*\frac{5}{2}}{7}$$

whereby "WE" represents weekend intake of nutrients and "WD" represents weekday intakes.

## Dietary intake analysis of the O-FFQ and the MY-VEG-FFQ

Using data from *Tzameret* [16] and the USDA database [17], we computed the nutrient content of the new food items that were added to the MY-VEG-FFQ list. The nutrient intake from both FFQs were calculated by multiplying the consumption frequency of each food item by its nutrient content per standard portion size, and then summing across all food items. Based on the distribution, participants with extreme values were filtered out (for women, energy >5000 kcal or <500 kcal; for men, >6000 kcal or <500 kcal). Moreover, to determine the value of the new questionnaire in improving assessment accuracy, we conducted comparisons between the consumption of food items and nutrients as reported by vegan participants who completed the MY-VEG-FFQ and those who had filled out the O-FFQ in the Tel Hai cohort.

## Sample size calculation

The G*Power software version 3.1.9 was used to calculate power analysis. The calculation was based on the correlation coefficients between the two measurement procedures. The minimum coefficient used was 0.4 with a power of 0.9. The minimum calculated sample size was N = 47. A review of FFQ validation studies found that the median sample size per study was 103, and that studies with sample sizes >103 tended to exhibit higher correlation coefficients than studies with smaller sample sizes [9]. Considering these findings, and due to financial and time limitations, we decided to recruit a minimum of 100 participants.

## Statistical analysis

Statistical analysis was conducted using the R statistical environment (version 4.2.2) and a range of libraries that extended the software's core functionalities. The analyses relied on the following R libraries: gtsummary, ggplot, ggpubr, and irr.

The f for all nutrients, food groups, and personal factor distribution was assessed using the Shapiro–Wilk test, which presented a clear deviation from normal for most variables. Consequently, most analyses were performed using nonparametric tests or natural log transformations to achieve normality.

A descriptive analysis was also performed, and the means and standard deviations of the daily consumption of each nutrient were calculated, followed by comparisons between the two dietary assessment methods (the MY-VEG-FFQ and the 3-DFR) using the Wilcoxon rank sum test (crude analysis). Moreover, the following three statistical methods were used to validate the novel FFQ in the three-day food records: (1) The Pearson correlation coefficients (crude and per 1000 calories) for log transformed values, whereby we considered a correlation efficiency of >0.4 to be meaningfully acceptable; (2) cross-classifications based on Cohen's quadratic weighted kappa values; and (3) The Bland-Altman method analysis (for estimating limits of agreement (LOA) assessments). Finally, to compare the MY-VEG FFQ and the O-FFQ, we used analysis of covariances (ANCOVA) following log transformation.

Results Of the 242 participants who signed the consent form to participate in this study, 197 completed the MY-VEG-FFQ, including 101 participants who also submitted their 3-DFRs for the validation portion of this research. Of these 197 participants, 136 were vegans and their input was used to conduct comparisons between the MY-VEG-FFQ and the O-FFQ (S2 Fig).

Most participants in the validation section of the study were female (83.2%), and 30.7% were overweight or obese. The average age was 38.7 years, and 91.1% held an academic or associate degree (i.e., >12 years of schooling). The characteristics of those who participated in the validation (n = 101), i.e., the vegan participants who completed the O-FFQ (n = 90), and those who completed the MY-VEG-FFQ (n = 136) are presented in Table 1.

Nutrient intake estimates from the 3-DFR and from the MY-VEG-FFQ are presented in Tables 2 and S1, with the former tending to be lower than the latter. The mean total energy intake was 1,722 (421) kcal and 1,912 (518) kcal per day for the 3-DFR and the

**Table 1. Study population characteristics by study group [a,b].**

| Characteristic | MY-VEG FFQ & 3-DFR N = 101 [a] | MY-VEG FFQ, N = 136 [b] | O-FFQ, N = 90 [b] | p-value [c] |
|---|---|---|---|---|
| Women (%) | 84 (83.2%) | 112 (82.4%) | 56 (62.2%) | <0.001 |
| Age (years) Mean (SD) | 38.7 (11.5) | 36.3 (11.5) | 32.5 (10.3) | 0.010 |
| Body mass (kg) Mean (SD) | 65.2 (15.9) | 66.6 (17.4) | 64.1 (12.2) | 0.6 |
| BMI (kg/Meter$^2$) Mean (SD) | 23.7 (5.2) | 24.1 (6.0) | 22.8 (3.4) | 0.2 |
| BMI in categories | | | | 0.3 |
| Underweight | 8 (7.9%) | 7 (5.1%) | 6 (6.7%) | |
| Normal | 62 (61.4%) | 85 (62.5%) | 64 (71.9%) | |
| Overweight | 24 (23.8%) | 32 (23.5%) | 13 (14.6%) | |
| Obese | 7 (6.9%) | 12 (8.8%) | 6 (6.7%) | |
| Smoking (%) | 16 (15.8%) | 30 (22.1%) | 10 (11.1%) | 0.035 |
| >12 School years (%) | 92 (91.1%) | 112 (82.4%) | 45 (50.0%) | <0.001 |
| Married or with partner (%) | 67 (66.3%) | 88 (64.7%) | 39 (43.3%) | 0.002 |

FFQ = food-frequency questionnaire; O-FFQ = original food-frequency questionnaire; 3-DFR = three-day food records; BMI = body mass index (kg/m$^2$); SD = standard deviation.

[a] Participated in the validation (completed the MY-VEG-FFQ & 3-DFR)

[b] Vegan participants who took part in the comparison analysis between the MY-VEG FFQ and O-FFQ

[c] Comparison between vegan participants who completed the MY-VEG FFQ and those who filled out the O-FFQ using Pearson's chi-squared test for categorial variables and Wilcoxon rank sum test for continuous variables.

**Table 2. Mean daily nutrient intakes, estimated by the MY-VEG-FFQ and the three-day food records.**

| Nutrient Intake (Units/day) | 3-DFR N = 101 | MY-VEG FFQ N = 101 | Diff. | 95% CI | p-value [a] |
|---|---|---|---|---|---|
| | Mean (SD) | Mean (SD) | | | |
| Energy (kcal) | 1,722 (421) | 1,912 (518) | -190 | -321, -59 | 0.005 |
| Carbohydrates (% E) | 49.2 (8.3) | 46.7 (6.7) | 2.5 | 0.4, 4.6 | 0.021 |
| Protein (% E) | 14.5 (3.2) | 15.0 (2.4) | -0.6 | -1.4, 0.2 | 0.15 |
| Total Fat (% E) | 36.4 (7.3) | 38.3 (5.8) | -1.9 | -3.7, -0.0 | 0.044 |
| Saturated (% E) | 8.0 (4.0) | 7.4 (2.2) | 0.6 | -0.3, 1.5 | 0.2 |
| Dietary Fiber (g) | 38.7 (14.8) | 46.5 (16.2) | -7.8 | -12.1, -3.5 | <0.001 |
| Calcium (mg) | 676.1 (273.3) | 839.1 (287.7) | -163.0 | -240.8, -85.1 | <0.001 |
| Iron (mg) | 15.5 (9.1) | 17.1 (5.6) | -1.6 | -3.7, 0.5 | 0.14 |
| Sodium (mg) | 2,632.2 (964.9) | 2,694.4 (835.8) | -62.1 | -312.6, 188.4 | 0.6 |
| Zinc (mg) | 8.5 (4.3) | 9.4 (3.0) | -0.9 | -1.9, 0.1 | 0.084 |
| Vitamin E (mg) | 10.3 (3.9) | 14.8 (5.1) | -4.5 | -5.7, -3.2 | <0.001 |
| Vitamin B9 (mcg) | 466.5 (278.8) | 573.6 (201.6) | -107.1 | -174.7, -39.6 | 0.002 |

3-DFR = three-day food records; CI = Confidence Interval; FFQ = Food-Frequency Questionnaire; Diff: Different

[a] Welch Two Sample t-test.

MY-VEG-FFQ, respectively (PV = 0.005). Protein intake was 59.4(19.4) and 68.6 (21.3) for both dietary assessment tools, respectively (PV = 0.001). The same patterns were observed for micro-nutrients. The percentage of energy from macro-nutrients was similar for protein and saturated fat, and the difference in the percentage of energy from carbohydrates was significant, albeit minimal (2.5%).

Calorie-adjusted Pearson correlation coefficients (S2 Table) ranged from r = 0.25 (for vitamin E) to 0.63 (for cholesterol). The results of the cross-classification analysis are presented in S3 Table. The proportion of participants with exact quartile agreement between the MY-VEG FFQ and the 3-DFR ranged between 29.7% (for zinc) and 47.5% (for total fat). Further analysis for the exact quartile or adjacent quartiles ranged from 73.3% (for phosphorus) to 82.1% (for carbohydrates). The Bland-Altman analysis validated that the MY-VEG-FFQ tended to overestimate the consumption of nutrient intake. LOA assessments using Bland-Altman methods were wide. For example, the LOA for energy was between 600–1,422 calories (S3 Fig). These results are comparable with other FFQ validation studies worldwide, including the Israeli O-FFQ (S4 Fig).

Tables 3 and S4 show comparisons between the dietary intake of vegans using the MY-VEG-FFQ and the O-FFQ. In the new FFQ, participants had a significantly higher intake of energy, protein (calculated in grams and energy percentage), total and saturated fat (calculated in grams and energy percentage), carbohydrates (estimated in energy percentage), dietary fiber, calcium, iron, phosphorus, sodium, zinc, and vitamin B9. No differences were seen in consumption of carbohydrates (in grams), potassium, vitamin E, vitamin C, vitamin B3, and vitamin B6. Cholesterol consumption was significantly lower in the MY-VEG-FFQ than in the O-FFQ. Following adjustments for age and sex, these differences remained statistically significant.

The vegan participants who completed the MY-VEG-FFQ reported consumption of a larger number of food items than vegans who completed the O-FFQ [65.5 (13.5) and 51.5 (12.5), respectively]. The contribution of new and/or modified food items to various nutrients, compared to identical food items on the O-FFQ is depicted in Fig 1. More than half the energy

**Table 3. Daily consumption of nutrients of vegans, estimated by the O-FFQ and the MY-VEG-FFQ.**

| Nutrient | MY-VEG-FFQ N = 136 | O-FFQ N = 90 | P value[a] | Adjusted p value [b] |
|---|---|---|---|---|
| | Mean (SD) | Mean (SD) | | |
| Food energy (kcal) | 1,963 (657) | 1,698 (823) | <0.001 | <0.001 |
| Carbohydrates (% E) | 46.8 (6.7) | 54.2 (9.5) | <0.001 | <0.001 |
| Protein (% E) | 15.2 (2.4) | 13.0 (2.9) | <0.001 | <0.001 |
| Total Fat (% E) | 38.0 (5.5) | 32.8 (7.9) | <0.001 | <0.001 |
| Saturated Fat (% E) | 6.8 (1.5) | 6.0 (1.9) | <0.001 | <0.001 |
| Dietary Fiber (g) | 48.5 (17.8) | 42.3 (21.2) | 0.001 | 0.001 |
| Cholesterol (mg) | 2.1 (8.5) | 35.2 (67.7) | <0.001 | <0.001 |
| Calcium (mg) | 823.7 (329.9) | 720.3 (336.5) | 0.005 | 0.011 |
| Iron (mg) | 18.2 (6.5) | 14.0 (7.2) | <0.001 | <0.001 |
| Sodium (mg) | 2,830.3 (1,098.7) | 2,538.6 (1,605.3) | 0.003 | <0.001 |
| Zinc (mg) | 9.5 (3.4) | 8.0 (3.7) | <0.001 | <0.001 |
| Vitamin E (mg) | 14.4 (5.2) | 14.7 (7.3) | 0.8 | >0.9 |
| Vitamin B9 (mcg) | 585.3 (219.8) | 515.9 (255.8) | 0.006 | 0.006 |

FFQ = Food-Frequency Questionnaire

[a] ANCOVA (log transformed)

[b] ANCOVA (log transformed) adjusted for sex and age.

consumed by the vegans who completed the MY-VEG-FFQ stemmed from new or modified food items. This contribution ranged from 18% for vitamin E to 36% for protein in modified food items; and ranged from 24% for carbohydrates to 44% for total fat. Finally, vegans who completed the new MY-VEG-FFQ reported higher consumption of baked goods, legumes,

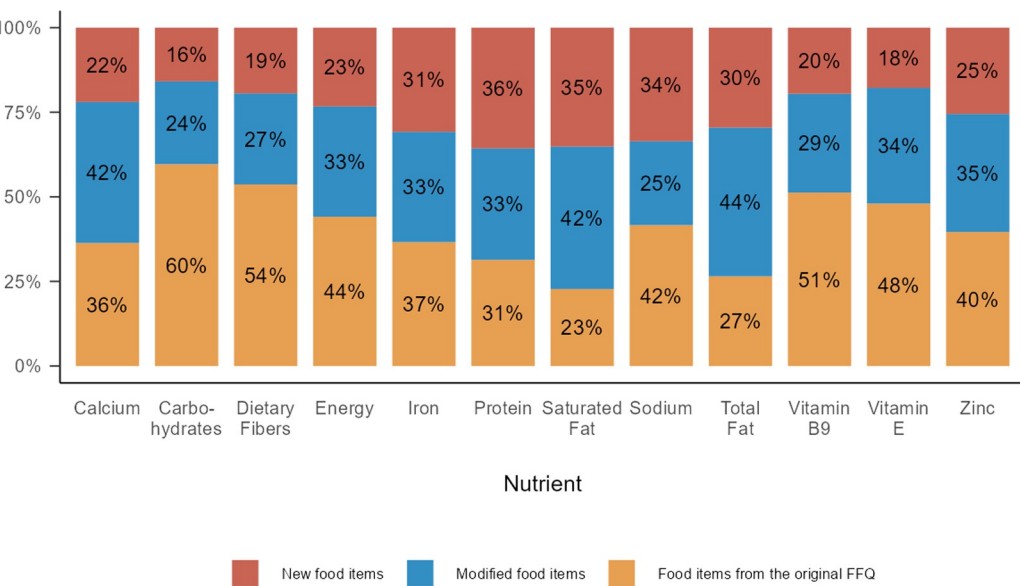

**Fig 1. Distribution of nutrient consumption sources by food items among vegans who completed the MY-VEG-FFQ.** Comparison between food items that appeared in the O-FFQ and new food items: New food items are those that were not on the O-FFQ (e.g., tofu-based dishes); modified food items were included in the O-FFQ, yet required minor adjustments (e.g., baked goods); food items from the O-FFQ that were included in the O-FFQ (e.g., rice and tomatoes). FFQ = food-frequency questionnaire; O-FFQ = Original FFQ.

meat and milk analogs, dishes that contain tofu and seitan, avocado, nuts, and seeds; however, they reported lower consumption of whole grains (S5 Table).

## Discussion

This study aimed at developing a modular web-based FFQ that is suitable for all types of diets in Israel, including those that are plant-based. To validate the new MY-VEG-FFQ, data collected from 202 participants for this questionnaire were compared to those gathered from 90 vegans who had completed the previously validated O-FFQ. For most nutrients, similar coefficients were seen in the new FFQ compared to previous studies. A recent meta-analysis, which assessed the validity of 67 FFQs that entailed food records as a reference method, found a range of 0.173–0.735 coefficients (median = 0.373) [9], which is, indeed, comparable to our results. Specifically, the crude and adjusted correlations were comparable to those reported by Dyett et al. [18] and Jaceldo-siegl et al. [19] in relation to two FFQs for vegans in the United States that were validated against repeated 24-hour recalls.

The crude analysis and Bland-Altman plot revealed that the FFQ tends to overestimate dietary intake when compared to the 3-DFR, suggesting limited accuracy in assessing individual dietary adequacy. This tendency toward overestimation is in line with a recent meta-analysis by Cui et al. [9] Nonetheless, similar to other FFQs, this questionnaire can still be used to rank the nutrient status of individuals, as supported by the findings of the cross-classification analysis presented in this study. The analysis demonstrates a reasonable level of agreement, with approximately 73.3%–82.1% of the participants accurately classified in the same quartile or an adjacent one when considering adjusted nutrient intake. Similar results have been reported in other recent studies [20–23].

Using a web-based FFQ offers a range of benefits, such as the elimination of participants skipping questions or choosing more than one answer for a certain question [12]. Additionally, such surveys enable standardized questioning, with increased consistency in the collected data, a decrease in missing data, and the elimination of errors due to manual data entries and coding. Additionally, participants are ensured increased confidentiality and are required to spend less time completing such questionnaires, as web-based surveys are faster and easier to complete than hard copies [24, 25]. Finally, using a skip algorithm tailored to the dietary habits of the participants can reduce participation burden and survey length, and increase responsiveness [25]. While our focus was on dietary habits related to animal-based foods, this modular approach can be useful for addressing or avoiding other food groups.

Vegans who filled out the MY-VEG-FFQ reported a larger number of food items than those who completed the O-FFQ. Additionally, the MY-VEG-FFQ showed a significantly higher intake of most macro and micro-nutrients. This could indicate that using an FFQ for the general population may underestimate the actual nutrient consumption of vegans. This was also seen in the nutritional contribution of O-FFQ items, which was less than 50% for most nutrients among vegans who completed MY-VEG-FFQ.

Nutritional deficiencies of protein, iron, calcium, zinc, and other minerals are a major concern in vegan diets [26–29]. Our findings suggest that inadequate dietary assessment methods contribute to this concern. It is crucial to use an adjusted FFQ to accurately evaluate nutrient intake in vegans. Failure to carefully adjust FFQs for this population may lead to significant reporting bias. Two recent reviews on nutrient consumption among vegans yielded conflicting results regarding energy and nutrient intake [26, 27]. While one review, which focused on vegans in European countries, found lower energy, protein, zinc, and calcium intake [26], another review of worldwide data found no differences in energy, zinc, and calcium intake, but did find lower protein intake among vegans [27]. While this may be due to genuine differences

between vegans in European countries and those in other places, it is important to assess how the nutrient intake was measured in different studies. Most studies in the European review used FFQs without specific adjustments for vegans, [26], whereas the worldwide review utilized more diverse dietary assessment tools. Only half of the studies employed FFQs, some of which were validated for vegan and vegetarian populations [27].

Such bias could help explain the disparity in nutrient intake observed between the EPIC-Oxford Study (EPOS) [30] and the Adventist Health Study 2 (AHS2), two very large-scale studies on vegetarians and vegans in high-income countries [31]. Both studies used carefully validated FFQs. However, while the AHS2 FFQ was validated for populations with high plant-based item intake [19], the EPOS FFQ was initially validated for the general population [32] and only later modified for vegans [30]. It is worth mentioning that no differences were seen in energy intake between omnivores and vegans in the AHS2 [33], whereas in the EPOS, vegans consumed fewer calories during the recruitment [30] and the follow-up stages [34]. While this difference may partly stem from genuine disparities between the cohorts, the underestimation issue with the EPOS FFQ cannot be disregarded.

In this study, vegans who filled out the O-FFQ reported an average cholesterol consumption of 35.2 mg per day compared to 2.1 mg among those who completed the MY-VEG-FFQ. As dietary cholesterol is limited to animal foods, this may imply false animal product consumption, which in turn, may have led to methodological problems when estimating such intake [35]. Compared to those who completed the O-FFQ, vegans who completed the MY-VEG-FFQ reported consuming more legumes, tofu and seitan, nuts and seeds, milk analogs, and meat analogs (including tofu and seitan). We believe that the higher reported consumption of these items stems from the number of options/lines in the new questionnaire, compared to the one-food item in the O-FFQ.

Specifically, the MY-VEG-FFQ can better assess different food patterns of vegetarians and vegans, as it separates ultra-processed meat analogs from traditional ones. This is becoming increasingly important, as the variety of ultra-processed plant-based food options continues to grow [6, 7]. Other studies on plant-based diets have shown that vegetarians and vegans tend to consume more of these food items than omnivores [5, 26], thereby emphasizing the importance of including various options for these food items on FFQs and gaining a better understanding of different food patterns and diet quality among these populations. In addition, our study found that vegans who completed the MY-VEG-FFQ ate more baked goods than those who completed the O-FFQ. This may be explained by the egg and dairy content in regular pastries compared to vegan baked foods.

## Research limitations

The authors of this study acknowledge the limitations associated with conducting a study of this nature. First, the participants were selected from a convenience sample, which may not adequately represent the target population and caution is needed in generalizing the findings to other populations. Moreover, most participants were female, held an academic degree, and had normal BMI measures. Yet, as validation studies entail within-participant comparisons, it is reasonable to assume that the selection bias did not affect the validity coefficients in this study. Moreover, while the final sample size used in the validation study was conservative (n = 101), there was agreement as to using samples with 100 participants or fewer for validation studies [36]. This number is also close to the median sample size (n = 103) found in a recent review of 130 FFQ validation studies [9]. In addition, the study's timeline and available resources enabled the collection of a maximum of three days of food records, based on self-reporting, and without the weighing of the reported consumed food items. Future studies

could benefit from gathering food records that cover a larger number of days, which could also help reduce the daily variation of infrequently consumed foods. Finally, stability over time may be affected, as only one entry was made per FFQ, and no reproducibility measures were taken due to financial and time constraints.

In addition, it is important to note that we used food records as our gold standard for the validation process, rather than biomarkers that are free of reporting biases. However, biomarkers also have disadvantages, as they do not reflect the amount of food components consumed–only the absorption of these components in the body, which is affected by their bioavailability in different foods. In addition, absorption is also influenced by individual physiological conditions and diseases. Furthermore, biological marker tests are relatively expensive and complicated to use, and they cannot be applied for every food component [36].

## Conclusions

The MY-VEG-FFQ was developed to address challenges encountered when attempting to tailor food lists in FFQs to dietary preferences, especially plant-based diets. The novel FFQ demonstrated reasonable validity for most nutrients, with comparable correlation coefficients to other FFQ validation studies. However, it tended to overestimate dietary intake compared to three-day food records, thereby limiting its accuracy in capturing precise nutrient consumption. Nonetheless, this FFQ is able to rank participants' nutrient status. Moreover, the MY-VEG-FFQ provided valuable insights into the dietary patterns and nutrient intake of both vegetarians and vegans, while highlighting their unique food choices. This web-based tool offers a practical and cost-effective method for assessing long-term dietary intake among plant-based individuals and comparing them to omnivores. It could be further refined and expanded to accommodate additional, newly emerging dietary patterns. Overall, the MY-VEG-FFQ contributes to improving dietary assessment accuracy and advancing research on the health and environmental implications of plant-based diets.

## Supporting information

**S1 Fig. Skip algorithm in the My-VEG-FFQ.**
(PDF)

**S2 Fig. Study flowchart.**
(PDF)

**S3 Fig. Bland-Altman plots of individual differences in nutrient intake between the My-Veg-FFQ and the three-day food record (based on averages).**
(PDF)

**S4 Fig. Comparison of correlations ($R^2$) for different dietary components between the current study and validation studies of FFQ questionnaires from previous studies.**
(PDF)

**S1 Table. Daily consumption of nutrients as estimated by the 3-DFR and MY-VEG-FFQ–crude comparison (full).**
(PDF)

**S2 Table. Correlations between the MY-VEG-FFQ and the three-day food record.**
(PDF)

**S3 Table. Cross-classification of quartiles of macro-nutrient intake derived from My-VEG-FFQ vs. three-day food record.**
(PDF)

**S4 Table. Daily consumption of nutrients estimated by the O-FFQ for omnivores and by the MY-VEG-FFQ for vegetarians and vegans.**
(PDF)

**S5 Table. Daily consumption of various food items estimated by the O-FFQ for omnivores and by the My-VEG-FFQ for vegetarians and vegans.**
(PDF)

**S1 File. FFQ in English.**
(PDF)

**S2 File. Raw data.**
(XLSX)

**S1 Checklist. Human participants research checklist.**
(PDF)

## Acknowledgments

We would like to acknowledge the research participants in the MY-FFQ-VEG study and in the Tel-Hai Cohort for their contribution to this study. We also wish to thank Beverley Yohanan for her professional editing of this article.

## Author Contributions

**Conceptualization:** Kerem Avital, Sigal Tepper, Sivan Ben-Avraham, Danit Rivka Shahar.

**Data curation:** Kerem Avital.

**Formal analysis:** Kerem Avital.

**Investigation:** Kerem Avital.

**Methodology:** Kerem Avital, Sigal Tepper, Danit Rivka Shahar.

**Project administration:** Kerem Avital, Danit Rivka Shahar.

**Resources:** Kerem Avital.

**Software:** Kerem Avital.

**Supervision:** Sigal Tepper, Danit Rivka Shahar.

**Validation:** Kerem Avital.

**Visualization:** Kerem Avital.

**Writing – original draft:** Kerem Avital.

**Writing – review & editing:** Sigal Tepper, Sivan Ben-Avraham, Danit Rivka Shahar.

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
