## [Decision Letter · Decision Letter 0]

4 Oct 2023

PONE-D-23-25091Development and validation of the MY-VEG-FFQ: A modular web-based food-frequency questionnaire for vegetarians and vegansPLOS ONE

Dear Dr. Shahar,

Thank you for submitting your manuscript to PLOS ONE. After careful consideration, we feel that it has merit but does not fully meet PLOS ONE’s publication criteria as it currently stands. Therefore, we invite you to submit a revised version of the manuscript that addresses the points raised during the review process.

We look forward to receiving your revised manuscript.

Kind regards,

**Diego A. Bonilla**

Academic Editor

PLOS ONE

Journal Requirements:

4. Please remove your figures from within your manuscript file, leaving only the individual TIFF/EPS image files, uploaded separately. These will be automatically included in the reviewers’ PDF.

Reviewers' comments:

Reviewer's Responses to Questions

**Comments to the Author**

1. Is the manuscript technically sound, and do the data support the conclusions?

Reviewer #1: Yes

Reviewer #2: Yes

2. Has the statistical analysis been performed appropriately and rigorously? 

Reviewer #1: Yes

Reviewer #2: Yes

3. Have the authors made all data underlying the findings in their manuscript fully available?

Reviewer #1: Yes

Reviewer #2: No

4. Is the manuscript presented in an intelligible fashion and written in standard English?

Reviewer #1: Yes

Reviewer #2: Yes

5. Review Comments to the Author

Reviewer #1: First I want to thank the editor for the opportunity to review this manuscript and then I want to congratulate the authors for their initiative and its quality.

Now, below I will present a series of comments that I hope can help improve the document.In the statistical analysis section there is no mention of the use or not of tests that evaluate the type of distribution of the data. Since the authors use statistics to analyze data with Gaussian and non-Gaussian distribution, it would be appropriate to state the reason for this situation.

Within the discussion and limitations of the study, no reference is made to the need or not to estimate the stability over time of the results. It is interesting to know if the authors consider that their instrument can provide stable and reproducible results over time or perhaps this is a limitation not addressed in the document.

While as stated in reference (36), a sample size around 100 subjects seems to be sufficient, I can't help but wonder why they didn't calculate their own sample size?

In Table 2, values appear in parentheses in the cells of the second and third column. What data are these? What contribution do these values make to the analysis of results?

Reviewer #2: The manuscript is notably intriguing as it introduces a fresh approach to nutritional control in vegan subjects, among other significant aspects.

However, it would be beneficial if the authors provided a more detailed explanation of the inclusion and exclusion criteria for participant subjects in the validation of the instrument. For example, were subjects with specific dietary restrictions excluded? Could subjects with any cognitive limitations (mild or otherwise) complete the questionnaire? Clarifying these aspects would help in better understanding the methodology used.

Furthermore, a thorough review of the tables is suggested, as at times they present figures with three decimal places, while in other instances, only two or even one decimal place is used. It would be advisable to standardize the format in accordance with the journal's style guidelines.

Regarding Table 2, the values presented in parentheses after the mean and standard deviation are not clear. It is recommended to add a note at the end of the table to clarify this aspect and provide a proper understanding of the data.

Another important point is the lack of consistency in table presentation. Some tables follow one format, while others have a different style. Improving this aspect to achieve a more consistent and professional presentation is suggested.

The authors should also consider providing complete information about the validated instrument or questionnaire, including the questions and response options. This would enable a better understanding of the tool used in the research.

The information presented at the end of Table 4, specifically the median, is not clear. The authors should review if there is any error in the calculation of this descriptive statistic and correct it as needed.

Furthermore, an excessive number of tables are observed in the manuscripts. It would be advisable to consider the possibility of consolidating information into one table or expressing some data in text form to simplify the presentation.

Finally, it is suggested that one of the graphs be eliminated, and the information it contains be presented in text form, which could enhance overall clarity and presentation.

6. PLOS authors have the option to publish the peer review history of their article (what does this mean?). If published, this will include your full peer review and any attached files.

Reviewer #1: **Yes: **Jhonatan González-Santamaría

Reviewer #2: No

---

## [Author Response · Author response to Decision Letter 0]

14 Nov 2023

Reviewer #1:

In the statistical analysis section there is no mention of the use or not of tests that evaluate the type of distribution of the data. Since the authors use statistics to analyze data with Gaussian and non-Gaussian distribution, it would be appropriate to state the reason for this situation.

We appreciate the reviewer's comment. In response, we have included additional details regarding the distribution tests conducted to determine the appropriate analysis method. (Lines 167-170) Furthermore, we have added information about the transformations applied to the data on nutrients and food groups before conducting Pearson correlation tests. (lines 176)

Additionally, the comment has prompted us to consider applying a logarithmic transformation before using the ANCOVA test in the adjusted comparison with the Original FFQ. We believe that this adjustment enhances the statistical robustness of the test. We are grateful for highlighting this important point, and we thank you once again for your valuable feedback. (lines 179-180, Table 3, S4, S5)

Within the discussion and limitations of the study, no reference is made to the need or not to estimate the stability over time of the results. 

It is interesting to know if the authors consider that their instrument can provide stable and reproducible results over time or perhaps this is a limitation not addressed in the document.

We agree with the reviewer and addressed the issue regarding the lack of reproducible results over time. (lines 349-350).

While as stated in reference (36), a sample size around 100 subjects seems to be sufficient, I can't help but wonder why they didn't calculate their own sample size?

We appreciate the reviewer's valuable input. We have now incorporated the sample size calculation and relevant considerations into the study method section (lines 181-188). Additionally, we have included a comparison to the sample sizes of 130 FFQ validation studies reviewed by Cui Q et al in the limitation section (lines 343-344).

In Table 2, values appear in parentheses in the cells of the second and third column. What data are these? What contribution do these values make to the analysis of results?

We appreciate the reviewer's comment. The values within the parentheses represent the median values. After careful consideration, we have decided to remove these values from table 2. This decision was made because the median values closely aligned with the mean values and did not significantly contribute to the results. This change also eliminates unnecessary complexity. Consequently, we have removed the median values from Tables 3, S1, S4, and S5 as well.

Reviewer #2: The manuscript is notably intriguing as it introduces a fresh approach to nutritional control in vegan subjects, among other significant aspects.

However, it would be beneficial if the authors provided a more detailed explanation of the inclusion and exclusion criteria for participant subjects in the validation of the instrument. For example, were subjects with specific dietary restrictions excluded? Could subjects with any cognitive limitations (mild or otherwise) complete the questionnaire? Clarifying these aspects would help in better understanding the methodology used.

We value the reviewer's comment and have included the inclusion and exclusion criteria for participants in the method section (lines 90-91). It is important to note that we did not specify any limitations related to cognitive impairments. As indicated in Table 1, the majority of the participants had more than 12 years of schooling and the mean age was 39 years old suggesting that cognitive tests are not required for this group. 

Furthermore, a thorough review of the tables is suggested, as at times they present figures with three decimal places, while in other instances, only two or even one decimal place is used. It would be advisable to standardize the format in accordance with the journal's style guidelines.

We have conducted a comprehensive review of the tables and have implemented the required modifications to maintain uniformity in the presentation of decimal places. Our approach aligns with that of Zheng et al, who published their research on the development and validation of a FFQ for children in PLOS One in 2020. In this approach, most means and percentage values are presented with one decimal place, except for energy values, which are displayed without decimal places. Correlations are depicted with two decimal places, following the article by Zheng et al [1] (Tables 1-3, S1-S5).

Regarding Table 2, the values presented in parentheses after the mean and standard deviation are not clear. It is recommended to add a note at the end of the table to clarify this aspect and provide a proper understanding of the data.

We appreciate the reviewer's comment. The values within the parentheses represent the median values. After careful consideration, we have decided to remove these values from table 2. This decision was made because the median values closely aligned with the mean values and did not significantly contribute to the results. This change also eliminates unnecessary complexity. Consequently, we have removed the median values from Tables 3, S1, S4, and S5 as well.

Another important point is the lack of consistency in table presentation. Some tables follow one format, while others have a different style. Improving this aspect to achieve a more consistent and professional presentation is suggested.

We have thoroughly reviewed the tables and have made the necessary adjustments to ensure consistency in the formatting of the tables. 

The authors should also consider providing complete information about the validated instrument or questionnaire, including the questions and response options. This would enable a better understanding of the tool used in the research.

Thank you for this suggestion, we added a supplement file with a translation of the FFQ used in our study. (File S1)

The information presented at the end of Table 4, specifically the median, is not clear. The authors should review if there is any error in the calculation of this descriptive statistic and correct it as needed. Furthermore, an excessive number of tables are observed in the manuscripts. It would be advisable to consider the possibility of consolidating information into one table or expressing some data in text form to simplify the presentation.

Following your suggestion, we merged table 4 and table S5. removed the median values, and improved the presentation of information in this table to enhance clarity.

Finally, it is suggested that one of the graphs be eliminated, and the information it contains be presented in text form, which could enhance overall clarity and presentation.

Following your suggestion, we removed figure 1 that presented the distribution of food items in the new and original FFQ.

---

## [Decision Letter · Decision Letter 1]

9 Jan 2024

PONE-D-23-25091R1Development and validation of the MY-VEG-FFQ: A modular web-based food-frequency questionnaire for vegetarians and vegansPLOS ONE

Dear Dr. Shahar,

Thank you for submitting your manuscript to PLOS ONE. After careful consideration, we feel that it has merit but does not fully meet PLOS ONE’s publication criteria as it currently stands. Therefore, we invite you to submit a revised version of the manuscript that addresses the points raised during the review process.

1. The English language requires improvement throughout the manuscript. For example, the authors state the word via 2 ties in this sentence: "The validation analysis showed that nutrient-intake estimates were generally higher via the MY-VEG-FFQ than via the three-day food records". Thorough proofreading by a native English speaker is recommended to enhance grammar (authors might attach the translation certificate for transparency and quality).

2. Although "body weight", "height" and "gender" are frequently used terms, it is technically correct to refer to "body mass", "stature" and "sex", respectively. Please address this accordingly through the manuscript.

3. Enhance the quality of the manuscript by following the STROBE guidelines for structuring the text and sections. For example, the subsection "sample size" should be before the "Statistical analysis". Attach the checklist accordingly, for example, https://www.strobe-statement.org/checklists/ - Cf, www.equator-network.org

4. Do not use "mean±standard deviation". Use Mean(SD) instead. Cf, PMID 21206631

5. Replace "Consequently, most analysis was computed using nonparametric tests or natural log transformations to increase normality" with "Consequently, most analyses were performed using nonparametric tests or natural log transformations to achieve normality."

6. The symbol of kilogram is kg not Kg. Revise units according to SI.

7. Be cautious with the interpretation and generalizability of the findings. 

We look forward to receiving your revised manuscript.

Kind regards,

Diego A. Bonilla

Academic Editor

PLOS ONE

Journal Requirements:

Reviewers' comments:

Reviewer's Responses to Questions

**Comments to the Author**

1. If the authors have adequately addressed your comments raised in a previous round of review and you feel that this manuscript is now acceptable for publication, you may indicate that here to bypass the “Comments to the Author” section, enter your conflict of interest statement in the “Confidential to Editor” section, and submit your "Accept" recommendation.

Reviewer #1: All comments have been addressed

Reviewer #2: All comments have been addressed

2. Is the manuscript technically sound, and do the data support the conclusions?

Reviewer #1: Yes

Reviewer #2: Yes

3. Has the statistical analysis been performed appropriately and rigorously? 

Reviewer #1: Yes

Reviewer #2: Yes

4. Have the authors made all data underlying the findings in their manuscript fully available?

Reviewer #1: Yes

Reviewer #2: Yes

5. Is the manuscript presented in an intelligible fashion and written in standard English?

Reviewer #1: Yes

Reviewer #2: Yes

6. Review Comments to the Author

Reviewer #1: Respected Editor

I have carefully read the authors' response to both my comments and those of the other reviewer.

I consider that the authors judiciously and detailed attention to each of the comments, they even made an effort to go beyond what the reviewers commented.

The effort and attention that the authors put into clarifying and adjusting the comments related to the statistical analysis and presentation of the data is undeniable. Likewise, they demonstrated openness regarding the presentation of additional information that was considered relevant.

Reviewer #2: Hey author, thanks for taking care of the reviewers' comments. That really helps to make sure our manuscripts are the best they can be.

7. PLOS authors have the option to publish the peer review history of their article (what does this mean?). If published, this will include your full peer review and any attached files.

Reviewer #1: **Yes: **Jhonatan Gonzalez Santamaría

Reviewer #2: **Yes: **Luis Alberto Cardozo

---

## [Author Response · Author response to Decision Letter 1]

9 Feb 2024

Point by point response for the editor:

1. The English language requires improvement throughout the manuscript. For example, the authors state the word via 2 ties in this sentence: "The validation analysis showed that nutrient-intake estimates were generally higher via the MY-VEG-FFQ than via the three-day food records". Thorough proofreading by a native English speaker is recommended to enhance grammar (authors might attach the translation certificate for transparency and quality).

The manuscript has undergone professional re-editing by the expert staff of American Manuscript Editors. Attached to this file, is their English editing certificate. We hope the re-edited version meets your expectations.

2. Although "body weight", "height" and "gender" are frequently used terms, it is technically correct to refer to "body mass", "stature" and "sex", respectively. Please address this accordingly through the manuscript.

We change the terms body weight", "height" and "gender” to body mass", "stature" and "sex", respectively (lines 94,232 and tables 1,3, S4, S5).

3. Enhance the quality of the manuscript by following the STROBE guidelines for structuring the text and sections. For example, the subsection "sample size" should be before the "Statistical analysis". Attach the checklist accordingly, for example, https://www.strobe-statement.org/checklists/ - Cf, www.equator-network.org

We implemented the required changes to adhere to the STROBE guidelines and shifted the subsection on sample size ahead of the statistical analysis (lines 163-170).

4. Do not use "mean±standard deviation". Use Mean(SD) instead. Cf, PMID 21206631

We convert the mean ± SD to mean (SD) throughout the manuscript, including in all the tables and supplements (lines 203,204, 240,241 and tables 1,2,3, S1, S4, S5)

5. Replace "Consequently, most analysis was computed using nonparametric tests or natural log transformations to increase normality" with "Consequently, most analyses were performed using nonparametric tests or natural log transformations to achieve normality."

We change the sentence as requested (lines 177,178)

6. The symbol of kilogram is kg not Kg. Revise units according to SI.

We change the units to kg (table 1)

7. Be cautious with the interpretation and generalizability of the findings. 

We add a remark regarding the interpretation and generalizability of the findings (line 339)

---

## [Editor Report · Decision Letter 2]

12 Feb 2024

Development and validation of the MY-VEG-FFQ: A modular web-based food-frequency questionnaire for vegetarians and vegans

PONE-D-23-25091R2

Dear Dr. Shahar,

We’re pleased to inform you that your manuscript has been judged scientifically suitable for publication and will be formally accepted for publication once it meets all outstanding technical requirements.

Kind regards,

**Prof. Diego A. Bonilla**

Academic Editor

PLOS ONE

Additional Editor Comments (optional):

Thanks for revising the manuscript, well done!

The manuscript is ready for publication, congratulations for this good contribution to literature.
---

## [Editor Report · Acceptance letter]

15 Feb 2024

PONE-D-23-25091R2 

PLOS ONE

Dear Dr. Shahar, 

I'm pleased to inform you that your manuscript has been deemed suitable for publication in PLOS ONE. Congratulations! Your manuscript is now being handed over to our production team.

Kind regards, 

on behalf of

Prof. Diego A. Bonilla 

Academic Editor

PLOS ONE